# Study on the Control of Saltwater Intrusion Using Subsurface Dams

Yawen Chang [1,2], Xuequn Chen [1,2,*], Dan Liu [1,2], Chanjuan Tian [1,2], Dandan Xu [1,2] and Luyao Wang [3]

1  Water Resources Research Institute of Shandong Province, Jinan 250013, China; skychangyawen@shandong.cn (Y.C.); skyliudan@shandong.cn (D.L.); skytianchanjuan@shandong.cn (C.T.); skyxudandan@shandong.cn (D.X.)
2  Shandong Provincial Key Laboratory of Water Resources and Environment, Jinan 250013, China
3  Shandong Green View Ecological Technology Company, Jinan 250013, China; 17775216650@163.com
*  Correspondence: skychenxuequn@shandong.cn; Tel.: +86-1358-3112-043

**Abstract:** Subsurface dams are widely used to prevent saltwater intrusion, with good results. This blockage often leads to an accumulation of pollutants and salt on the inland and seaside of the dam, respectively. While the latter is intended, the former effect is not desired and poses a huge problem in groundwater management. In order to quantitatively address this issue and clarify the impact of subsurface dam height, location, and the head difference for the saltwater and freshwater boundary on saltwater wedges and fresh groundwater discharge, a flow tank and numerical model were constructed. The results indicate that there was an optimal effective dam height and location (also the minimum effective dam location) for controlling saltwater intrusion, which corresponded to the maximum groundwater and freshwater discharge. When the various conditions of the numerical model were set according to the flow tank and the dam was 15 cm away from the saltwater boundary, the minimum effective dam height was equal to the aquifer thickness multiplied by 0.36. The dam height reached a height that was slightly higher than the minimum effective height, namely, the ratio of dam height to aquifer thickness was 0.38, which revealed that the freshwater discharge reached its maximum at 22.71 $cm^3/min$, the saline water wedge area was the smallest at 378 $cm^2$, and the prevention effect of saltwater intrusion was the best. Building a dam too high, that was, the ratio of dam height to aquifer thickness exceeded 0.38, resulted in an increased saltwater wedge area and exacerbated aquifer pollution. When the dam was located at the minimum effective distance for preventing saltwater intrusion under a certain dam height and head difference between saltwater and freshwater boundary, that was, the ratio of the distance of the dam to the saltwater boundary to the total length of the aquifer was 0.063, the distance of the dam to the saltwater boundary was the minimum effective distance. Compared to other effective distances, when the dam was at the minimum effective distance, the freshwater discharge reached its maximum at 22.71 $cm^3/min$, and the saltwater wedge area was the smallest at 378 $cm^2$. These conclusions provide a theoretical reference for the impact of subsurface dam construction on the saltwater wedge. This study examines the impact of tides and waves on the water head of the saltwater boundary, and it is also necessary to verify these conclusions through actual field experiments. We will investigate this in future work.

**Keywords:** saltwater intrusion; subsurface dam; flow tank; saltwater wedge; dam heights; hydraulic gradient

## 1. Introduction

Seawater intrusion is a natural phenomenon that is driven by the density difference between freshwater and seawater, and it occurs when the salt–freshwater interface moves landward to achieve equilibrium [1,2]. However, the development of local economies, climate change, and other external factors, as well as the overexploitation of groundwater, changes in land-use types, and rising sea levels in coastal areas have damaged the dynamic balance of the original salt–freshwater interface, intensifying the degree of seawater

intrusion [3–7]. Global changes in seawater intrusion have led to decreased freshwater resources and the abandonment of wells in coastal areas, restricting the economic development of coastal areas [8–12]. Currently, some recent techniques are being used for mapping saltwater intrusions. Mohamed Abdelfattah et al. [13] used various integrated approaches including geophysical survey, field investigations, well drilling, well logging, pumping tests, and water sampling to perform a detailed analysis of the hydrogeological and hydrochemical characteristics of a coastal aquifer in the West Port Said area, northeastern Egypt, and assess the desalination suitability of the aquifer. Finally, they discovered the most suitable aquifer for desalination. Ian Gottschalk et al. [14] investigated the extent of, and controls on, saltwater intrusion using the AEM method in the northern Salinas Valley, CA, USA. The findings demonstrated the value of acquiring AEM data for investigating the distribution of salinity in coastal aquifers impacted by saltwater intrusion.

Currently, several methods, such as the artificial recharge of groundwater or the establishment of subsurface dams, have been proposed to address the issue of seawater intrusion [12–23]. Onder et al. [24] detailed the types, design, and construction techniques of subsurface dams and suggested dams as a means of sustainable development. Numerical simulations using MODFLOW have also been used to evaluate the effectiveness of subsurface dams in preventing seawater intrusion and analyze the impacts of dams on groundwater flow fields. The results indicate that subsurface dams can increase the effective storage capacity of aquifers and effectively prevent seawater intrusion, allowing the sustainable utilization of water resources. Yuan Yirang et al. [25] simulated a subsurface dam using the moisture-proof dike and engineering regulation modes for the prevention and control of seawater intrusion in the Laizhou Bay area, with results showing that dams placed both upstream and downstream in the seawater intrusion area had a significant effect, with the depth and length of the downstream dam directly affecting the outcome. Tsanis and Song [26] used SUTRA to numerically simulate seawater intrusion and remediation in the Upper Florida Aquifer of South Carolina and concluded that installing injection wells somewhat prevents seawater intrusion. However, these studies only focus on confined aquifers. Ebeling et al. [27] established a two-dimensional variable-density seawater intrusion model using SEAWAT and FloPy to study the feasibility and optimal management strategy of a mixed hydraulic barrier scheme, which was then compared with a single pumping hydraulic barrier. Abdoulhalik et al. [28] proposed a hybrid physical barrier system that combines an impermeable cutoff wall with a semi-permeable underground dam as a new type of barrier system for controlling seawater intrusion. This system uses freshwater to drive saltwater upward toward the coastline, significantly reducing the intrusion range of saltwater bodies in coastal aquifers. The effectiveness of the anti-seepage walls in preventing the intrusion of seawater into multiphase aquifers was also examined.

Hydraulic barriers can achieve good results in preventing saltwater intrusion over short time periods; however, problems with well pipe blockage occur over longer periods [29,30]. Therefore, hydraulic barriers have not been widely applied for the prevention of saltwater intrusion [31]. Physical barriers, such as subsurface dams or cutoff walls, are costly; however, these methods can prevent saltwater intrusion on a permanent basis with superior effectiveness compared to hydraulic barriers [25,32].

Extensive research has been conducted to investigate the changes in the salt–freshwater interface resulting from pumping/injection wells [3,33–37]. Although subsurface dams are important for groundwater management in coastal areas, research investigating the effects of subsurface dam height, location, and the head difference between saltwater and freshwater boundaries on saltwater wedges and fresh groundwater discharge is limited. Li Fulin et al. [38] constructed a flow tank and numerical model to simulate the process of saltwater intrusion. They used fitted numerical models to observe saltwater intrusion by changing the dam's permeability. These research results indicated that the subsurface barriers with low permeability ($K = 3.7 \times 10^{-8}$ m/s) could prevent the saltwater intrusion effectively. Although this study investigated the impact of subsurface dam permeability on saltwater intrusion, it did not investigate the effects of subsurface dam height, location, and

the head difference between the saltwater and freshwater boundary on saltwater intrusion. Luyun et al. [39] studied the relationship between the height of subsurface dams and the thickness of the saltwater wedge using laboratory tests and SEAWAT. They found that, when the subsurface dams were higher than the thickness of the saltwater wedge, seawater intrusion could be prevented, and the saltwater trapped upstream could be flushed out. However, the effects of the construction location of the subsurface dams and the head difference were neglected. Vassilios K. Kaleris et al. [40] simulated the impact of different dam heights and locations on the saltwater wedge toe length. The above studies only use the saltwater wedge toe length as a criterion for evaluating saltwater intrusion. We used the fresh groundwater discharge to assess the environmental performance of the subsurface dam, as an increased fresh groundwater discharge is beneficial for carrying land-based pollutants and salt to the sea, which accumulate in traditional high subsurface barriers. In this study, a flow tank was used to compare saltwater intrusion without intervention with that following the construction of the subsurface dams. Based on the relevant flow tank parameters, a numerical model was built, corrected, and used to simulate the development of saltwater intrusion and freshwater discharge under different dam heights, locations, and hydraulic gradients in order to optimize the dam height and location under different scenarios.

## 2. Materials and Methods

### 2.1. Experimental Methods

#### 2.1.1. Laboratory Materials

A sand tank, saltwater tank, and freshwater tank were used to produce the experimental flow tank. Transparent organic glass was used to construct the $1.6 \times 0.1 \times 0.5$ m sand tank (Figure 1), rendering the simulated sand visible. Milky white quartz filtered through a 40–80 mesh sieve was used as sand. The connection between the sand tank and freshwater tank and between the sand tank and saltwater tank was constructed using a porous organic glass plate with a filter screen adhered to the inside and a nonporous acrylic plate inserted to ensure that the side walls of the sand tank were watertight. A valve was connected to the freshwater circulation device via a rubber pipe inserted into the right side of the freshwater tank, and a similar valve was used for the saltwater tank. A water head of 43.5 cm was maintained in the freshwater tank and a 42.5 cm one was used in the saltwater tank. Tap water was used as freshwater and a 36 g/L NaCl solution dyed with carmine was utilized to simulate saltwater.

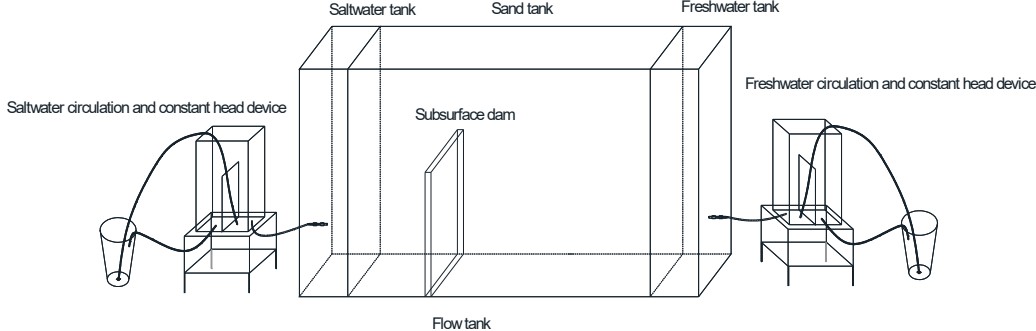

**Figure 1.** Schematic of the experimental device.

Parameters such as conductivity, dispersion coefficient, and porosity were examined, with results suggesting conductivity, longitudinal dispersion coefficient, and porosity of $8.7 \times 10^{-2}$ cm/s, 0.52 cm, and 0.30, respectively.

#### 2.1.2. Experimental Setup

Quartz sand was slowly added to the bottom of the flow tank, and pressure was added every 10 cm using an acrylic board. Water was gradually added to prevent the dispersal

of the medium. Water addition ceased when the water level in the sand tank reached 43.5 cm and was maintained. A 36 g/L NaCl solution was prepared, and carmine dye was added for use in the saltwater constant head device. Freshwater was maintained at 43.5 cm and saltwater with a concentration of 36 g/L at 42.5 cm. The valves on both the freshwater and the saltwater side were then connected to the constant head device. Then, the laboratory experiment was started to simulate the saltwater intrusion in the natural state. After the saltwater intrusion simulation in the nature state was completed, the water body was emptied from the flow tank and an acrylic board with a height of 33 cm was inserted into the flow tank 30 cm from the saltwater boundary to simulate dam interference. The same experimental steps were used for dam conditions.

*2.2. Numerical Simulation*

SEAWAT-2000 was used to establish a numerical model of groundwater flow and solute transport that could simulate the process of saline water intrusion. It was then coupled with MT3D and MODFLOW-2000 to consider the effect of density on groundwater flow. SEAWAT-2000 has been widely used in the calculation of variable-density processes, such as seawater intrusion [41], brine exploitation, and submarine groundwater discharge [42,43].

2.2.1. Governing Equation

The governing equation for groundwater flow can be expressed as:

$$\frac{\partial}{\partial x}\left(\rho K_{fx}\left[\frac{\partial h_f}{\partial x} + \left(\frac{\rho - \rho_f}{\rho_f}\right)\frac{\partial Z}{\partial x}\right]\right) + \frac{\partial}{\partial y}\left(\rho K_{fy}\left[\frac{\partial h_f}{\partial y} + \left(\frac{\rho - \rho_f}{\rho_f}\right)\frac{\partial Z}{\partial y}\right]\right) + \frac{\partial}{\partial z}\left(\rho K_{fz}\left[\frac{\partial h_f}{\partial z} + \left(\frac{\rho - \rho_f}{\rho_f}\right)\frac{\partial Z}{\partial z}\right]\right)$$
$$= \rho S_f \frac{\partial h}{\partial t} + \theta \frac{\partial \rho}{\partial C}\frac{\partial C}{\partial t} - \overline{\rho_s} q_s \tag{1}$$

where $h_f$ is the equivalent freshwater head [L]; $\rho_f$ is the density of freshwater [ML$^{-3}$]; $q_s$ is the unit volume flow of the source (sink) [T$^{-1}$]; $\theta$ is the effective porosity of the porous medium; $S_f$ is the unit water storage coefficient of equivalent freshwater [L$^{-1}$]; $K_f$ is the permeability coefficient of equivalent freshwater [LT$^{-1}$].

The solute transport equation is expressed using:

$$\frac{\partial\left(\theta C^k\right)}{\partial t} = \frac{\partial}{\partial x_i}\left(\theta D_{ij}\frac{\partial C^k}{\partial x_i}\right) - \frac{\partial}{\partial x}\left(\theta \vartheta_i C^k\right) + q_s C_k^s + R_n \tag{2}$$

where $C_k$ is the dissolved concentration of substance $k$ [ML$^{-1}$]; $D_{ij}$ is the hydrodynamic dispersion coefficient [L$^2$T$^{-1}$]; $C_k^s$ is the concentration of substance $k$ in source or sink [ML$^{-1}$]; $R_n$ is the reaction term for the utilized chemical substance.

2.2.2. Numerical Model

A numerical model was constructed based on various flow tank parameters (Figure 1) with a homogeneous aquifer used to simulate natural conditions. The simulation was performed over the range of 160 × 10 × 50 cm, with the area divided into 41 layers × 160 columns × 1 line, totaling 6560 cells. A stress period was added hourly, and the simulation was performed over a total of 2000 h. The initial water level, boundary conditions, dispersion coefficient, and permeability coefficient were input into the model as the main parameters, and the initial groundwater level of the model was set at 43.5 cm. The bottom of the model was set to an impermeable boundary and the top freewater surface boundary. The left side was set to the constant head and concentration boundary with a groundwater head of 42.5 cm and a chloride ion concentration of 36 g/L, while the right side was set to the constant head and concentration boundary with a groundwater head of 43.5 cm and a salt concentration of 0 g/L.

*2.3. Evaluation Parameters*

The correlation coefficients that were used to evaluate the impact of a subsurface dam on saltwater intrusion mainly included the following: $h/H$ (subsurface dam height divided by the thickness of the aquifer, H); $h_{salt}$ (height of the saltwater wedge in the location of the subsurface dam); $Q_0$ (freshwater discharge at the freshwater boundary), $\Delta d$ (the difference between the freshwater head $d_2$ and the saline water head $d_1$), $L/L_{Total}$ (distance $L$ from the location of the subsurface dam to the saltwater boundary divided by the total length of the aquifer, $L_{Total}$), $L_{Salt}$ (saltwater wedge toe length), and $A$ (area of the saltwater wedge). The saltwater wedge was set at a 50% concentration, which was commonly used to describe saltwater wedges in aquifers with low dispersivity [44].

## 3. Results and Discussion

### 3.1. Simulation Results and Calibration

Figure 2a,b shows the saltwater wedge that occurs naturally when solute transport reaches a steady state, as obtained from laboratory experiments and numerical simulations. The observed and calculated lengths obtained for the saltwater wedge toe over time are shown in Figure 2c and indicate that the solute transport reached an equilibrium state after 280 min, with a maximum variation of 79 cm obtained for the saltwater wedge toe length. The mean error (ME) of 3 cm is less than 10% of the variation range, indicating that the simulation matched the observation well.

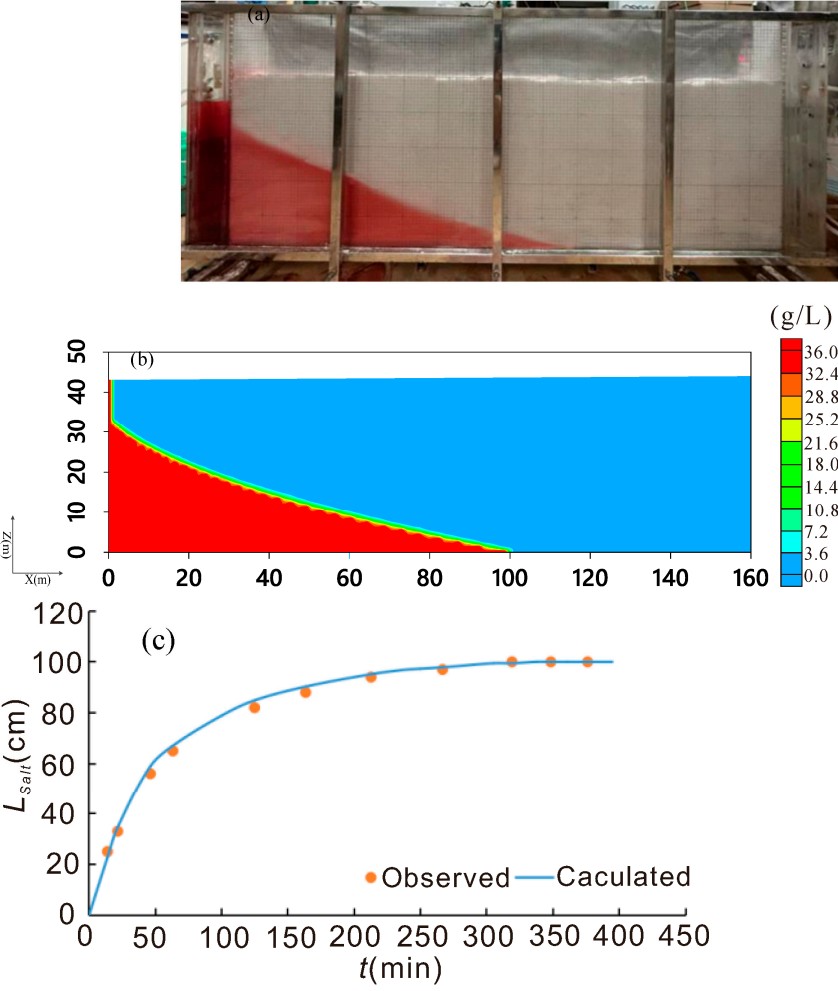

**Figure 2.** Distribution of saltwater wedge obtained through a (**a**) laboratory experiment and (**b**) numerical simulation, and (**c**) the observed and calculated values of the saltwater wedge toe length over time.

Figure 3a,b shows the acrylic board with a height of 33 cm that was inserted 30 cm from the saltwater boundary at the location of the second support from the left of the flow tank and the distribution of the saltwater wedge in the flow tank when the solute reached the equilibrium state 80 min after the start of the experiment. Figure 3c shows the saltwater wedge toe length at the end of the simulation period from both observation and calculation. The maximum variation in the saltwater wedge toe length within the flow tank reached 30 cm with an ME of 2 cm, which is less than 10% of the variation range. The simulated saltwater wedge matches the observation well.

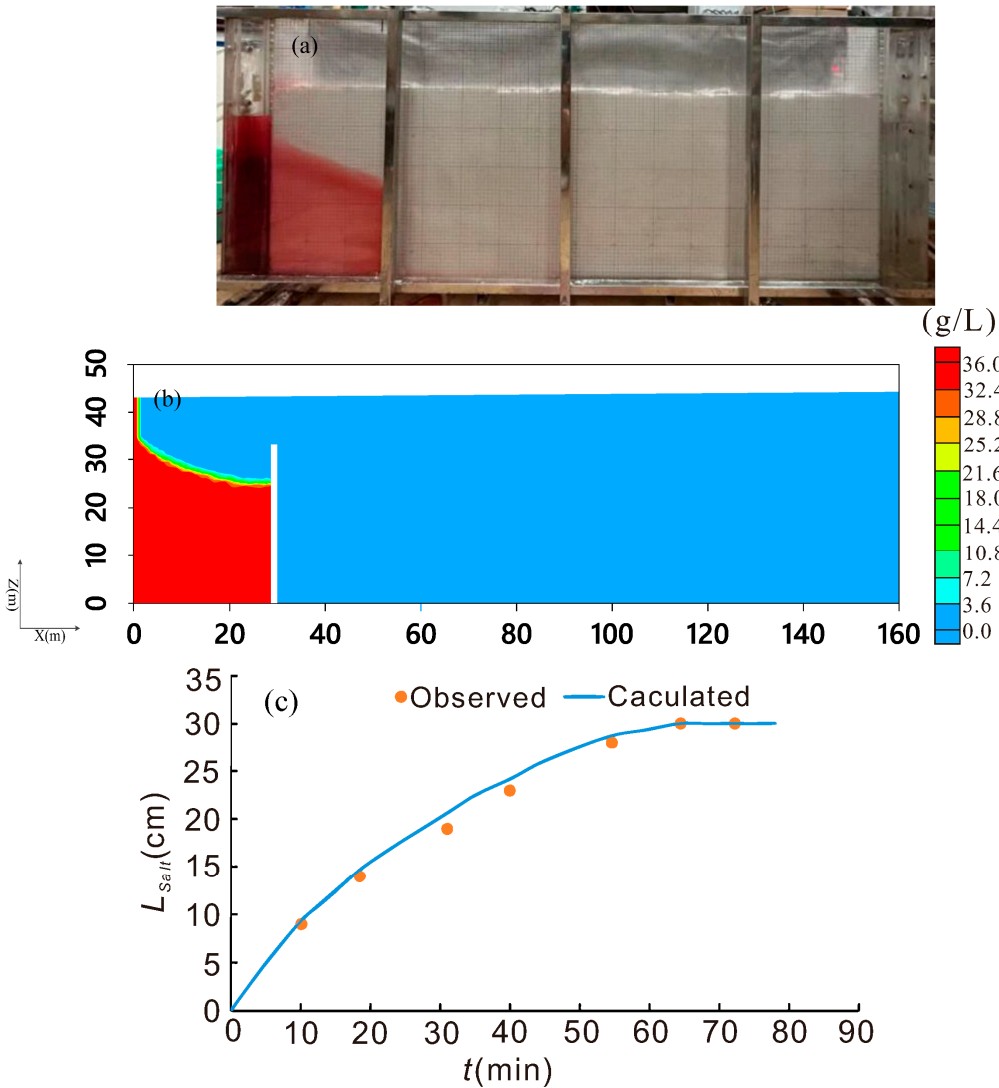

**Figure 3.** Distribution of saltwater wedge after subsurface dam was set through a (**a**) laboratory experiment. (**b**) Numerical simulation. (**c**) Observed and calculated values of saltwater wedge toe length over time.

Overall, the numerical model was found to be effective and could, therefore, be used for analysis and research.

### 3.2. Analysis of the Control Effect of Subsurface Dams on Saltwater Intrusion

To analyze how dams control saltwater intrusion, the dam height, distance, and head difference between the saltwater and freshwater boundaries were changed in the calibrated numerical model, allowing a better understanding of how these factors affect saltwater intrusion.

### 3.2.1. Influence of Dam Height

A subsurface dam was constructed 15 cm from the salt boundary, and the relationship between the salt wedge toe length and the height of the subsurface dam was analyzed. When the head difference was 13 mm between the saltwater and freshwater (with the saltwater boundary head at 43 cm and the freshwater boundary head at 44.3 cm), the salt wedge toe length was 101.1 cm and the saltwater wedge height was 24 cm at 15 cm from the saltwater boundary. The dam height of 16 cm allowed the saltwater wedge to flow over the top, and the saltwater wedge toe length reached 100.065 cm when the model reached a steady state. At a dam height of 17 cm, the saltwater wedge toe length reached 99.46 cm upon reaching a steady state, indicating that the dam could not effectively intercept the saltwater at a height of 16 or 17 cm because the dam top was located below the saltwater wedge, allowing saltwater to intrude. At dam heights of 16 or 17 cm, a steady state was reached 340 and 480 min after initiation, respectively, which was 26% and 78% longer than the 270 min required for equilibrium to be attained without intervention. These results indicate that increasing the height of a dam can delay intrusion.

Figure 4a shows the quantitative relationship between the height of a dam located 15 cm from the salt boundary and the saltwater wedge length. Beneath an $h/H$ of 0.36 (at $h = 18$ cm), as long as sufficient time was allowed, the length of the seawater wedge was similar to that observed without intervention, indicating that the dam has no effect on preventing seawater intrusion. The seawater wedge was completely intercepted at an $h/H$ of 0.36, indicating that $h/H = 0.36$ can be considered the minimum height, $h_{min}$, at which a dam can effectively prevent saltwater intrusion. At heights less than $h_{min}$, increasing the dam height had no significant effect on reducing saltwater intrusion; however, at dam heights that were equal to or higher than $h_{min}$, saltwater intrusion can be effectively intercepted.

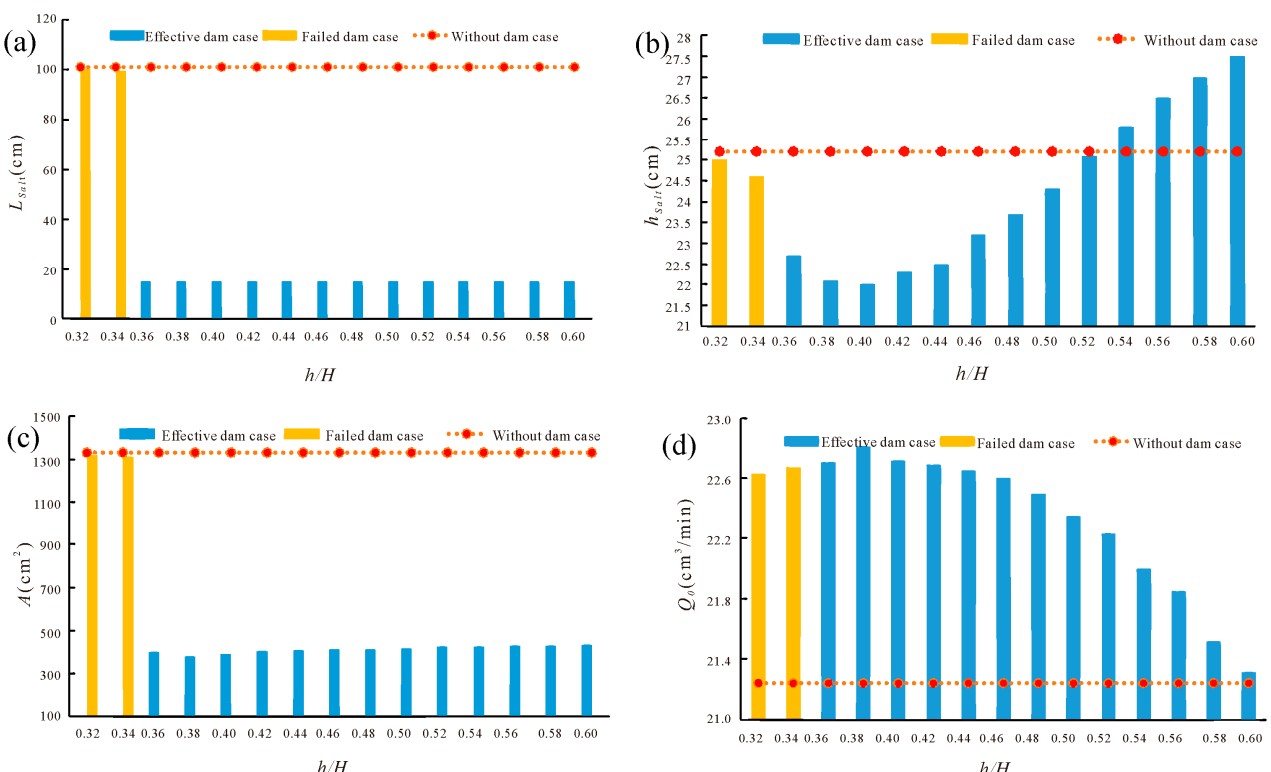

**Figure 4.** Relationship between dam height ($h/H$) and (**a**) saltwater wedge toe length ($L_{Salt}$), (**b**) saltwater wedge height at the location of the dam ($h_{Salt}$), (**c**) saltwater wedge area ($A$), and (**d**) freshwater discharge ($Q_0$).

Figure 4b,c shows the relationship between the dam height when located 15 cm away from the salt boundary and the height and area of the saltwater wedge. The saltwater

wedge was only intercepted by the dam if the dam height was increased to 18 cm, with the saltwater wedge being positioned slightly higher than the dam and lower than the saltwater wedge height that would occur without intervention. However, saltwater wedge heights of 22.7 and 22.1 cm were observed, respectively, at dam heights of 18 and 19 cm, with corresponding saltwater wedge areas of 404 and 378 cm$^2$, respectively. At dam heights of 20, 21, and 27 cm, saltwater wedge heights of 22, 22.3, and 25.8 cm were observed, respectively, corresponding to areas of 389, 406, and 429 cm$^2$, respectively. It can be seen that, at dam heights greater than 27 cm, the saltwater wedge height at the dam exceeds the saltwater wedge height achieved without intervention, with both the saltwater wedge height and area increasing alongside the dam height. Therefore, dams that are too high lead to seawater pollution in the aquifer, and dams that are only slightly higher than hmin are associated with the best prevention and saltwater intrusion control.

As shown in Figure 4d, the freshwater discharge first increased and then decreased as the dam height increased. At $h/H = 0.38$ ($h = 19$ cm), the freshwater discharge reached the maximum value of 22.71 cm$^3$/min, which is higher than that observed without intervention (21.24 cm$^3$/min). However, the saltwater wedge area was the smallest (Figure 4c). These results suggest that the change in freshwater discharge is related to the size of the saltwater wedge area under the influence of the dam, and that, the smaller the saltwater wedge area, the greater the freshwater discharge.

### 3.2.2. Influence of the Distance between Dam and Saltwater Boundary

Figure 5 shows the simulation results obtained when the distance from the dam to the saltwater boundary was varied under a certain dam height and groundwater head difference between the saltwater and freshwater boundary. Figure 5a,b shows the simulation results obtained when the dam was 6 and 8 cm away from the saltwater boundary, with a head distance of 13 mm between the saltwater and freshwater boundary (with the saltwater boundary head at 43 cm and the freshwater boundary head at 44.3 cm), and the dam height was set to 20 cm. When the dam was 6 cm away from the saltwater boundary, the saltwater wedge passed through the top of the dam and reached a steady state, with the saltwater wedge length reaching 100.42 cm. When the dam was 8 cm away from the saltwater boundary, the saltwater wedge passed through the top of the dam and reached a steady state, with the saltwater wedge length reaching 99.89 cm. The effects of the dam's location were investigated by placing the dam 10, 13, 15, and 25 cm away from the saltwater boundary, and the results suggest a role in the prevention of saltwater intrusion (Figure 5c–f).

Figure 6a shows the relationship between the dam position and saltwater wedge length when the head difference was set at 13 mm (with the saltwater boundary head at 43 cm and the freshwater boundary head at 44.3 cm) and the dam height at 20 cm. The results show that, up to an $L/L_{Total}$ ratio of <0.063 ($L = 10$ cm), the saltwater wedge toe length slowly decreased from 100.92 cm to 99.47 cm as $L/L_{Total}$ increased. At $L/L_{Total} = 0.063$, the saltwater wedge toe length decreased rapidly to 10 cm, and the dam effectively prevented saltwater intrusion. As $L/L_{Total}$ increased, the saltwater wedge toe length gradually increased until it reached the saltwater wedge toe length observed without intervention. Therefore, a minimum effective distance ($L_{min}$) of $L/L_{Total} = 0.063$ was obtained for the dam to saltwater when the head difference was 13 mm and the dam height was set at 20 cm.

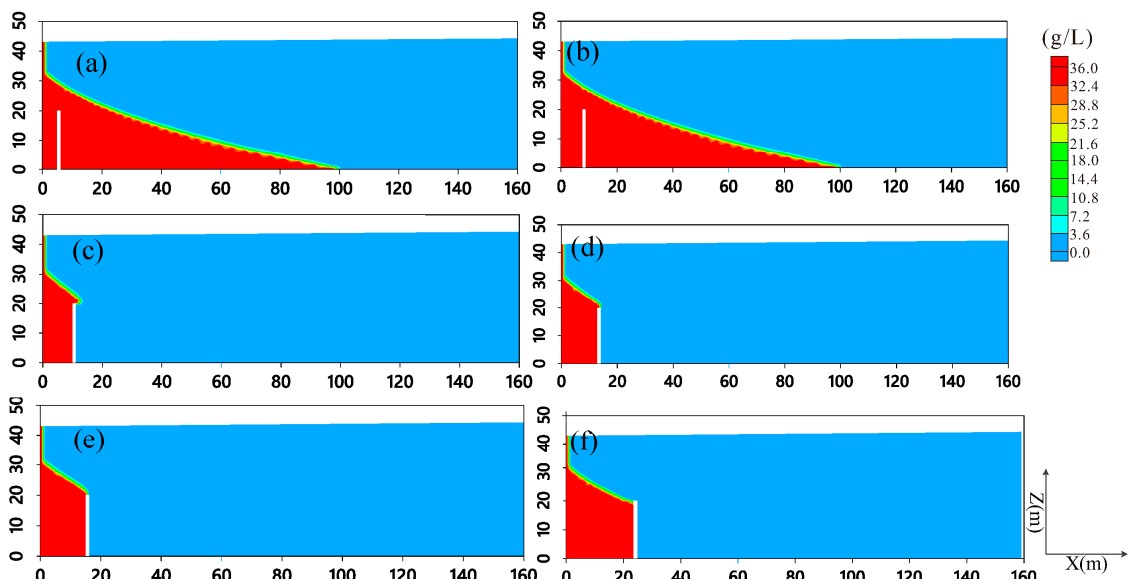

**Figure 5.** Saltwater wedge distribution obtained with (**a**) 6 cm, (**b**) 8 cm, (**c**) 10 cm, (**d**) 13 cm, (**e**) 15 cm, and (**f**) 25 cm between dam and saltwater boundary.

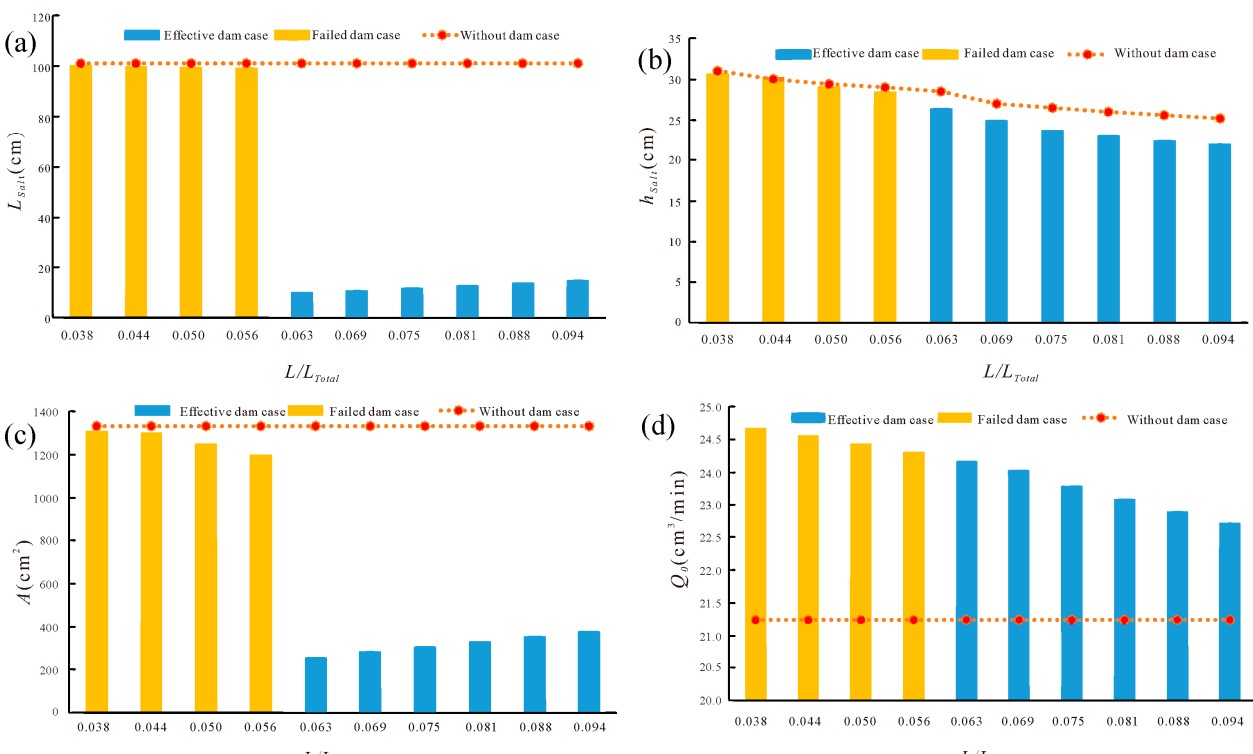

**Figure 6.** Relationship between dam position ($L/L_{Total}$) and (**a**) saltwater wedge toe length ($L_{Salt}$), (**b**) saltwater wedge height at dam ($h_{Salt}$), (**c**) saltwater wedge area ($A$), and (**d**) freshwater discharge ($Q_0$).

Figure 6b shows the relationship between the dam position and the saltwater wedge area when the head difference is 13 mm (with the saltwater boundary head at 43 cm and the freshwater boundary head at 44.3 cm) and the dam height is set to 20 cm. It was apparent that, when the dam position did not reach the $L_{min}$ required to block the saltwater, the saltwater wedge area was close to that observed without intervention. If the dam was at $L_{min}$, then the saltwater wedge area increased with the distance between the dam and the

saltwater boundary until it reached that observed without intervention. As the distance between the dam and the saltwater boundary increased, the saltwater wedge height at the location of the dam gradually decreased (Figure 7c), meaning that the effective dam height gradually decreased. Therefore, the dam's effective height gradually decreased. It is recommended to build a subsurface dam at a distance from the saltwater boundary as this reduces engineering costs but increases the saltwater wedge area.

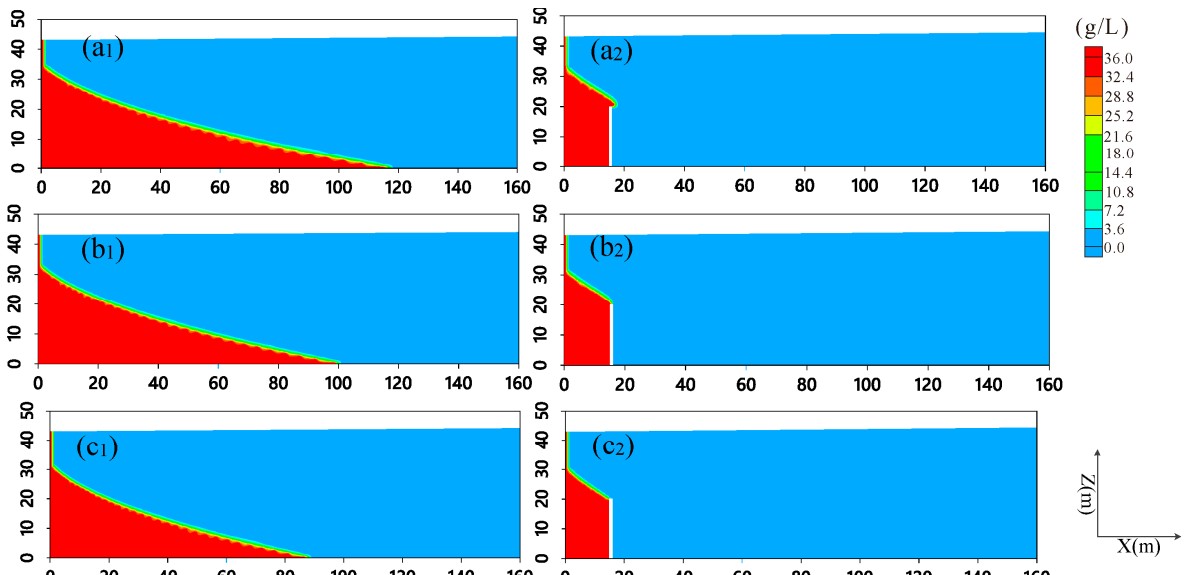

**Figure 7.** Saltwater wedge distribution. Hydraulic gradients of (**a₁**) 12 cm without intervention; (**a₂**) 12 cm with a dam; (**b₁**) 13 cm without intervention; (**b₂**) 13 cm with a dam; (**c₁**) 14 cm without intervention; (**c₂**) 14 cm with a dam.

As shown in Figure 6d, the amount of freshwater discharge gradually decreases as the distance between the dam and the saltwater boundary increases. $L/L_{Total}$ increased from 0.038 to 0.094, and freshwater discharge ($Q_0$) decreased from 24.17 cm³/min to 22.72 cm³/min, which was higher than without intervention (21.24 cm³/min). This is because the saltwater wedge area gradually increases and the freshwater discharge gradually decreases.

### 3.2.3. Influence of the Hydraulic Gradient

Different hydraulic gradients produced different results. Greater hydraulic gradients lead to shorter and lower salt wedges (Figure 7(a₁,b₁,c₁)), both with and without the presence of a dam (Figure 7(a₂,b₂,c₂)). Thus, increasing the hydraulic gradient improves the prevention and control of saltwater intrusion.

Figure 8 shows the results of the quantitative study investigating the impact of the hydraulic gradient on the saltwater wedge height at the dam, area, and length of the saltwater wedge, and freshwater discharge both without intervention and in the presence of a dam. To investigate the effects of changing the head between the saltwater and freshwater boundary, the dam height was set to 20 cm, the distance from the saltwater boundary was set to 15 cm, and the difference in the head between the saltwater and freshwater boundary was set to 10 mm (with the saltwater boundary head at 43.3 cm and the freshwater boundary head at 44.3 cm), 11 mm (with the saltwater boundary head at 43.2 cm and the freshwater boundary head at 44.3 cm), 12 mm (with the saltwater boundary head at 43.1 cm and the freshwater boundary head at 44.3 cm), 13 mm (with the saltwater boundary head at 43 cm and the freshwater boundary head at 44.3 cm), 14 mm (with the saltwater boundary head at 42.9 cm and the freshwater boundary head at 44.3 cm), and 15 mm (with the saltwater boundary head at 42.8 cm and the freshwater boundary head at 44.3 cm). Figure 8a shows the saltwater wedge toe length under the different groundwater heads. Without intervention, the saltwater wedge toe length decreased as the hydraulic

gradient increased in an approximately linear manner. However, in the presence of a dam, the saltwater wedge toe length gradually decreased as the head difference increased at head differences of <12 mm. However, the toe length decreased sharply to a distance consistent with the distance from the dam to the saltwater boundary when the head difference reached 12 mm. At head differences of >12 mm, the saltwater wedge was still intercepted at the dam site. Figure 8b,c shows the changes in the area and height of the saltwater wedge under different groundwater heads, with the area and height of the saltwater wedge decreasing gradually both without intervention and in the presence of a dam as the groundwater head difference increases. Figure 8d shows the changes in the freshwater discharged under different head differences, with the freshwater discharge increasing with the increased head difference. When the head difference was increased from 10 mm to 15 mm in the presence of a dam, the freshwater discharge increased from 15.07 $cm^3$/min to 27.23 $cm^3$/min, while the freshwater discharge increased from 12.93 $cm^3$/min to 26.3 $cm^3$/min without intervention. The freshwater discharge gradually decreased both without intervention and in the presence of a dam when the groundwater head difference increased.

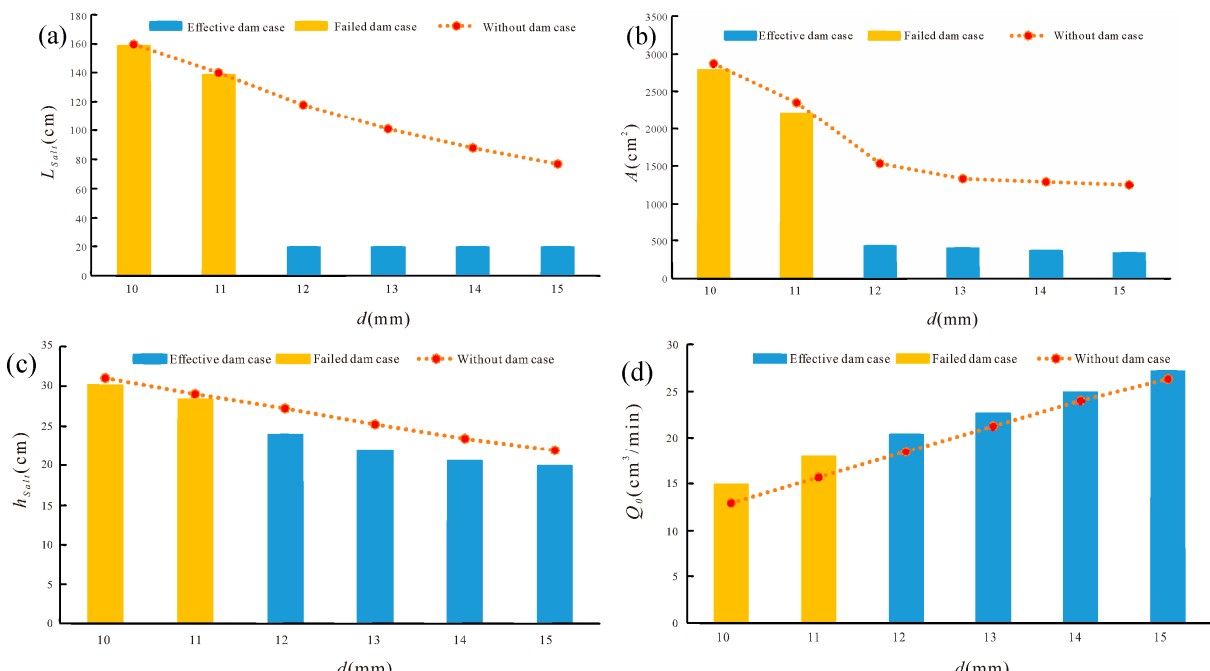

**Figure 8.** Relationship between hydraulic gradient ($\Delta d$) and (**a**) saltwater wedge toe length ($L_{Salt}$), (**b**) saltwater wedge area ($A$), (**c**) saltwater wedge height on dam location ($h_{Salt}$), and (**d**) freshwater discharge ($Q_0$).

## 4. Comparisons with Previous Studies

In previous studies, researchers conducted flow tank experiments to simulate the impact of subsurface dams on saltwater intrusion. Then, a two-dimensional numerical model was constructed based on the flow tank, and the calibrated numerical model was used to simulate the impact of subsurface dams on saltwater intrusion in various scenarios.

Li Fulin et al. [38] built a physical model and the FEFLOW-based groundwater numerical model to understand the dynamic processes of saltwater intrusion with and without subsurface barriers. When the subsurface barriers were induced, the hydraulic conductivity of barriers was lower, and the interception effect of the subsurface barriers was more significant. The subsurface barriers could effectively prevent the intrusion of saltwater when the permeability coefficient was less than $10^{-8}$ m/s. Luyun et al. [39] studied the relationship between the height of subsurface dams and the thickness of the saltwater wedge using laboratory tests and SEAWAT. They found that, when the subsurface dams were higher than the thickness of the saltwater wedge, seawater intrusion could be prevented,

and the saltwater trapped upstream could be flushed out. Luyun et al. [29] presented laboratory-scale investigations on how the effectiveness of cutoff walls varies with their depth and distance from the coast. The investigation showed that, when a cutoff wall is located in an area of saltwater intrusion, its protective effect increases with decreasing distance from the coast and increasing penetration depth. Antoifi Abdoulhalik et al. [28] completed numerical and laboratory experiments in a laboratory-scale aquifer in which the effectiveness of cutoff walls was assessed in three different configurations. The results show that the cutoff wall was effective in reducing the saltwater wedge in all the investigated cases of layered aquifers with a toe length reduction of up to 43%. The differences between previous research and this study are as follows:

(1)　Difference in whether the impact of subsurface dam location and height on the prevention and control of saltwater intrusion has been considered simultaneously.

Li Fulin et al. [38] did not consider the impact of subsurface dam location and height on saltwater intrusion when studying the impact of subsurface dam permeability on saltwater intrusion. In fact, when a subsurface dam's height or position is changed, the subsurface dam permeability that can effectively prevent saltwater intrusion may also change accordingly. Luyun et al. [39] considered the influence of subsurface dam height and thickness on saltwater intrusion but ignored the impact of subsurface dam location on saltwater intrusion. When the subsurface dam location is changed, the effectiveness of subsurface dam height and thickness in preventing saltwater intrusion may also change accordingly. Unlike previous studies, this study considers the impact of both subsurface dam location and height on the prevention and control of saltwater intrusion.

(2)　Difference in whether freshwater discharge is used as a criterion for evaluating the effectiveness of subsurface dams in preventing saltwater intrusion.

Previous studies [28,29,38,39] have used the saltwater wedge toe length change caused by the construction of a subsurface dam as a criterion for evaluating the effectiveness of subsurface dams in preventing saltwater intrusion. We used fresh groundwater discharge to assess the environmental performance of the subsurface dam. Although the saltwater wedge toe length is sometimes small, the subsurface dam height is relatively high; therefore, the subsurface dam blocks the groundwater movement both from and towards the sea. This blockage often leads to an accumulation of pollutants and salt on the inland and seaside of the dam, respectively. While the latter is intended, the former effect is not desired and poses a huge problem in groundwater management. Our study also found a relationship between dam height, fresh groundwater discharge, and saltwater wedge area when changing the dam height. When the fresh groundwater discharge reaches its maximum, the saltwater wedge area is the smallest, which indicates that the corresponding dam height is the optimal height for preventing seawater intrusion when the fresh groundwater discharge reaches its maximum.

## 5. Conclusions

The height, area of the saltwater wedge, and freshwater discharge following the installation of a dam have been neglected for some time, although the installation of a dam may lead to the severe salinization of the aquifer.

(1)　When the dam cannot effectively intercept saltwater, increasing the height can still delay saltwater intrusion. For a dam to have a preventive effect, it must reach the minimum effective dam height; increasing the height of a dam below this limit had no significant impact on reducing the saltwater wedge area, whereas the dam can effectively intercept saltwater intrusion if a dam is equal to or higher than the minimum effective height. However, dams that were far above this height were associated with an increase in the saltwater wedge area, exacerbating saltwater pollution. When the dam was slightly higher than the minimum effective height, the prevention and control effect of saltwater intrusion was the best. The change in freshwater discharge

was related to the size of the saltwater wedge area under the influence of the dam. The smaller the saltwater wedge area, the greater the amount of freshwater discharge was;

(2) Under a certain dam height and head difference between the saltwater and freshwater boundary, there was also a minimum effective distance for a dam to prevent saltwater intrusion. If the minimum effective distance was not achieved, the saltwater wedge area was close to the area under the natural state. If the minimum effective distance was achieved, the saltwater wedge area increased with the distance until it reached the natural state. The freshwater discharge gradually decreased as the distance between the dam and the saltwater boundary increased, as did the minimum effective height of the dam, reducing engineering costs but increasing the saltwater wedge area;

(3) The greater the hydraulic gradient, the shorter and lower the saltwater wedge, both in the presence and absence of a dam. Without intervention, the saltwater wedge toe length decreased as the hydraulic gradient increased in an approximately linear fashion. The freshwater discharge increased gradually as the hydraulic gradient increased, and the freshwater discharge gradually decreased as the head difference increased, both with and without intervention. Therefore, a large head difference can play a positive role in the prevention and control of saltwater intrusion, thereby reducing the construction costs of the dam.

Careful consideration should be given to the lowest local groundwater head difference (the difference between the lowest groundwater level and the maximum tidal height) to determine the minimum effective dam height when constructing an underground dam. An appropriate distance should then be determined based on the dam height.

The limitations of this study are as follows: (1) Because the scale of this study was on a laboratory scale, this study lacks the exploration of real-world geological and hydrogeological conditions in the field and lacks the inclusion of large-scale numerical models based on the investigated conditions and calibration of the model. The influence of the location and height of underground dams on the area and toe angle of saline water wedges, as well as on the discharge of fresh groundwater, also needs to be studied based on a calibrated model. (2) Our empirical experience with these setups hints at the fact that the temporal fluctuations of seawater heads due to tides and waves may have a significant impact on the elevation of a saltwater wedge. In this study, the saltwater boundary was set as a constant head boundary, without considering the impact of tides and waves on the water head of the saltwater boundary.

**Author Contributions:** Y.C.: conceptualization, methodology, investigation, writing—original draft preparation, formal analysis, funding acquisition; X.C.: supervision, conceptualization, methodology, formal analysis, resources, data curation, writing—reviewing and editing, project administration, funding acquisition; D.L.: methodology, resources; C.T.: investigation; D.X.: resources, data curation; L.W.: investigation. All authors have read and agreed to the published version of the manuscript.

**Funding:** This work was partly supported by Shandong Provincial Natural Science Foundation (Grant ZR2021MD086), Optional Project of Water Resources Research Institute of Shandong Province (Grant SDSKYZX202101), Shandong Provincial Natural Science Foundation (Grant ZR2022QD032), Key Technology and Application Project of Flood Control and Waterlogging Control in Plain Waterlogging Depression (Grant PCTXPQ-KY202001-2).

**Data Availability Statement:** The data generated and/or analyzed in the current study are not publicly available due to further dissertation writing but are available from the corresponding authors upon reasonable request.

**Conflicts of Interest:** Luyao Wang was employed by Shandong Green View Ecological Technology Company. The remaining authors declare that the research was conducted in the absence of any commercial or financial relationships that could be construed as a potential conflict of interest.

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
