# Peer review of "Study on the Control of Saltwater Intrusion Using Subsurface Dams"

_water, doi:10.3390/w15223938_

Round 1

Reviewer 1 Report

Comments and Suggestions for Authors

Chang et al manuscript review

The manuscript "Study on the control of saltwater intrusion by subsurface dams" needs significant revisions before it can be considered for publication. Although the manuscript is well-written, it lacks structure and requires expansion. Abstract, introduction, and Conclusions need to be improved to highlight the following points:

1.      The abstract needs to quantify numbers rather than using stretchy terms such as minimum, best … etc.

2.      This study relies solely on lab experiments and does not consider any real case studies for validation.

3.      In the introduction, the authors claim that limited similar work was conducted prior to this but they failed to summarize previous work outcomes.

4.      Authors need to compare their work with similar previous work.

5.      Conclusions need to highlight the limitations of this work

Please see the attached file for comments.

Author Response

Dear editor and Reviewer,

Thank you for your letter and for reviewer’s comments concerning our manuscript entitled “Study on the control of saltwater intrusion by subsurface dams”. Those comments are all valuable and very helpful for revising and improving our paper, as well as the important guiding significance to our researches. We have studied comments carefully and have made correction which we hope meet with approval. The amendments are marked up using the “Track Changes” function in the revised manuscript. Point by point responses to the reviewers’ comments are listed below this letter.

Comments and Suggestions for Authors

The manuscript "Study on the control of saltwater intrusion by subsurface dams" needs significant revisions before it can be considered for publication. Although the manuscript is well-written, it lacks structure and requires expansion. Abstract, introduction, and Conclusions need to be improved to highlight the following points:

1.The abstract needs to quantify numbers rather than using stretchy terms such as minimum, best … etc.

Authors response: We have quantified numbers in the abstract. Please see Page 1 Lines 19-33.  

2.This study relies solely on lab experiments and does not consider any real case studies for validation.

Authors response: Yes. This study can provide theoretical references for the behavior of the freshwater-seawater interface after the construction of subsurface dams. But this study lacks real case studies for validation, and it is the limitation of this work. We mentioned the limitation in the abstract and conclusion.Please see Page 1 Lines 34-36, and Page 14 Lines 483-493.

In the next work, we will explore actual geological and hydrogeological conditions in the field, and build large-scale numerical models and three-dimensional numerical models based on the investigated conditions. Using the calibrated model to study the impact of subsurface dams location, height, and other factors on the area and salt water wedge toe length caused by salt water intrusion, as well as the impact on fresh groundwater discharge.

3.In the introduction, the authors claim that limited similar work was conducted prior to this but they failed to summarize previous work outcomes.

Authors response: We have added a summary of previous work outcomes in the introduction. Please see Page 2-3 Lines 97-115.

4.Authors need to compare their work with similar previous work.

Authors response: We have added a section: comparisons with previous studies. Please see 4.Comparisons with previous studies (Page 12 Lines 338-446).

5.Conclusions need to highlight the limitations of this work.

Authors response: The limitations of this study are following. (1) Due to the scale of this study was laboratory scale, this study lacks the exploration of actual geological and hydrogeological conditions in the field, and lacks building large-scale numerical models based on the investigated conditions and the calibration of the model. The influence of the location and height of underground dams on the area and toe angle of saline water wedges, as well as on the discharge of fresh groundwater, also needs to be studied based on a calibrated model. (2) Our empirical experience with these setups hints on the fact that temporal fluctuations of seawater levels due to tides and waves may have a significant impact on the elevation of saltwater wedge. In this study, the saltwater boundary was set as a constant head boundary, without considering the impact of tides and waves on the water head of saltwater boundary.

Please see Page 14 Lines 483-493.

6.Please mention about the recent techniques used for mapping saltwater intrusions.

Authors response: We have added some recent techniques used for mapping saltwater intrusions. Please see Page 2 Lines 49-58.

7.Please mention references here and highlight the difference between what was done and what your are proposing.

Authors response: We have added references and highlight the difference between what was done and what our are proposing. Please see Page 2-3 Lines 97-115.

Reviewer 2 Report

Comments and Suggestions for Authors

1. The abstract is too lengthy and the focus is not prominent enough. The authors did not clarify the innovation of the manuscript and the challenge of the problem.

2. The figures are not clear enough, such as Figures 2c and 3c; The variables in the picture should be italicized.

3. Images cannot be captured using screenshots, but should be drawn and saved as vector images using software such as MATLAB and Visio.

4. the innovation or contribution is not very sufficient.

Comments on the Quality of English Language

OK

Author Response

Dear editor and Reviewer,

Thank you for your letter and for reviewer’s comments concerning our manuscript entitled “Study on the control of saltwater intrusion by subsurface dams”. Those comments are all valuable and very helpful for revising and improving our paper, as well as the important guiding significance to our researches. We have studied comments carefully and have made correction which we hope meet with approval. The amendments are marked up using the “Track Changes” function in the revised manuscript. Point by point responses to the reviewers’ comments are listed below this letter.

Comments and Suggestions for Authors

1.The abstract is too lengthy and the focus is not prominent enough. The authors did not clarify the innovation of the manuscript and the challenge of the problem.

Authors response: We have revised the abstract. The revised abstract focus on the  impact of subsurface dam height, location, and the head difference for the saltwater and freshwater boundary on saltwater wedges and fresh groundwater discharge, and used freshwater discharge as a criterion for evaluating the effectiveness of subsurface dam in preventing saltwater intrusion. These are also the innovation of the manuscript.  We have mentioned the challenge of the problem in the abstract. The challenge is that this study regards the impact of tides and waves on the water head of saltwater boundary,and it is also necessary to verify these conclusions through actual field experiments..We will investigate it in future work. Please see Page 1 Lines 10-36.

2.The figures are not clear enough, such as Figures 2c and 3c; The variables in the picture should be italicized.

Authors response: The figures have been drawn and saved as vector images, and the variables in the pictures have been revised to Italic. 

3.Images cannot be captured using screenshots, but should be drawn and saved as vector images using software such as MATLAB and Visio.

Authors response: The images have been drawn and saved as vector images.

4.The innovation or contribution is not very sufficient.

Authors response: We have added a section to compare their work with similar previous work, that is, 4. comparisons with previous studies.

The differences between previous research and this study are as follows:

  • Difference in whether the impact of subsurface dam location and height on the prevention and control of salt water intrusion has been considered simultaneously.
  • Difference in whether freshwater discharge is used as a criterion for evaluating the effectiveness of subsurface dam in preventing saltwater intrusion.

Please see Page 12-13 Lines 388-446.

Reviewer 3 Report

Comments and Suggestions for Authors

In the presented work, the Authors raised the important problem of protecting groundwater against mixing with salt water. The issue is important due to the need to protect groundwater resources that can be used to supply the population. The results of the presented research may help in designing safeguards to protect coastal groundwater resources.

The paper presents the results of laboratory tests and numerical modeling using SEAWAT. The assumptions and research methods are correctly selected, and the experimental results are correctly presented. Similar studies have already been presented in the scientific literature, the reviewed work is not new. Therefore, it should be stated that the work does not demonstrate the highest originality, although it concerns an important issue.

Author Response

Dear editor and Reviewer,

Thank you for your letter and for reviewer’s comments concerning our manuscript entitled “Study on the control of saltwater intrusion by subsurface dams”. Those comments are all valuable and very helpful for revising and improving our paper, as well as the important guiding significance to our researches. We have studied comments carefully and have made correction which we hope meet with approval. The amendments are marked up using the “Track Changes” function in the revised manuscript. Point by point responses to the reviewers’ comments are listed below this letter.

Comments and Suggestions for Authors

In the presented work, the Authors raised the important problem of protecting groundwater against mixing with salt water. The issue is important due to the need to protect groundwater resources that can be used to supply the population. The results of the presented research may help in designing safeguards to protect coastal groundwater resources.

The paper presents the results of laboratory tests and numerical modeling using SEAWAT. The assumptions and research methods are correctly selected, and the experimental results are correctly presented. Similar studies have already been presented in the scientific literature, the reviewed work is not new. Therefore, it should be stated that the work does not demonstrate the highest originality, although it concerns an important issue.

Authors response: We have added a section to compare their work with similar previous work, that is, 4. comparisons with previous studies.

The differences between previous research and this study are as follows:

  • Difference in whether the impact of subsurface dam location and height on the prevention and control of salt water intrusion has been considered simultaneously.
  • Difference in whether freshwater discharge is used as a criterion for evaluating the effectiveness of subsurface dam in preventing saltwater intrusion.

Please see Page 12-13 Lines 388-446.

Round 2

Reviewer 1 Report

Comments and Suggestions for Authors

The authors have addressed my comments.

Reviewer 2 Report

Comments and Suggestions for Authors

Accept